# Topoisomerase I (TOP1) dynamics: conformational transition from open to closed states

Diane T. Takahashi [1,2,3✉], Danièle Gadelle[1], Keli Agama[4], Evgeny Kiselev[4], Hongliang Zhang[4], Emilie Yab[2], Stephanie Petrella[2], Patrick Forterre[1], Yves Pommier [4✉] & Claudine Mayer[2,5,6]

Eukaryotic topoisomerases I (TOP1) are ubiquitous enzymes removing DNA torsional stress. However, there is little data concerning the three-dimensional structure of TOP1 in the absence of DNA, nor how the DNA molecule can enter/exit its closed conformation. Here, we solved the structure of thermostable archaeal *Caldiarchaeum subterraneum* CsTOP1 in an apo-form. The enzyme displays an open conformation resulting from one substantial rotation between the capping (CAP) and the catalytic (CAT) modules. The junction between these two modules is a five-residue loop, the hinge, whose flexibility permits the opening/closing of the enzyme and the entry of DNA. We identified a highly conserved tyrosine near the hinge as mediating the transition from the open to closed conformation upon DNA binding. Directed mutagenesis confirmed the importance of the hinge flexibility, and linked the enzyme dynamics with sensitivity to camptothecin, a TOP1 inhibitor targeting the TOP1 enzyme catalytic site in the closed conformation.

[1] Institut de Biologie Integrative de la Cellule, CNRS, Université Paris-Saclay, 91198 Gif sur Yvette, Cedex, France. [2] Institut Pasteur, Université de Paris, CNRS UMR 3528, Unité de Microbiologie Structurale, F-75015 Paris, France. [3] Biotechnology and Cell Signaling (CNRS/Université de Strasbourg, UMR 7242), Ecole Superieure de Biotechnologie de Strasbourg, Boulevard Sébastien Brant, BP 10413, F-67412 Illkirch, France. [4] Laboratory of Molecular Pharmacology, Developmental Therapeutics Branch, Center for Cancer Research, NCI, NIH, Bethesda, MD 20892, USA. [5] Université de Paris, F-75013 Paris, France. [6] Present address: ICube-UMR7357, CSTB, Centre de Recherche en Biomédecine de Strasbourg, 67084 Strasbourg, France. ✉email: dtakahashi@unistra.fr; pommier@nih.gov

D ue to the double-helical structure of DNA, its unwinding generates torsion during the translocation of replication and transcription complexes. The accumulation of torsional stresses leads to DNA supercoiling, which can block the progression of both machineries and requires the relaxase activity of DNA topoisomerases[1,2]. In addition, negative and positive supercoilings facilitate or inhibit DNA strands separation, respectively, and consequently aid or impair replication and transcription initiation[3]. As a consequence, the regulation of DNA topological state is particularly important for the regulation of gene expression and replication[4].

Based on their sequence comparison, DNA topoisomerases can be divided into five evolutionary distinct families—IA, IB, IC, IIA, and IIB—with different folds and reaction mechanisms[5]. The diversity of the topoisomerases reflects the specialization of the enzymes to particular DNA transactions, and also the diversity of organisms who evolved a different strategy to handle topological problems.

Type IB topoisomerases (Topo IB) are found in all eukaryotes (where they are referred to as TOP1), in several archaeal phyla, several bacterial genera and several groups of viruses[6]. TOP1, also known as swivelase or untwisting enzyme, can relax both positive and negative supercoils by nicking DNA, enabling the rotation of the broken DNA strand around the complementary intact strand of the DNA duplex and catalyzing the closing of the nick. This activity is particularly important during transcription to remove supercoils generated behind and ahead of the transcription bubble. In human, TOP1 has been shown to be functionally and physically associated with the transcription machinery[7]. Its molecular and cellular biology has been extensively studied because human TOP1 protein is the target of the widely used anticancer drugs, topotecan, and irinotecan, which are derived from the alkaloid camptothecin[8,9]. These drugs are known to specifically enter the catalytic site of TOP1 while it is covalently linked to the 3′-end of DNA, inhibiting the DNA rejoining step and dissociation of TOP1 from the end of the broken DNA[10]. The accumulation of TOP1-covalent complexes that are converted into double-strand breaks during replication is toxic, killing preferentially cancer cells, which often overexpress TOP1[11].

Human TOP1 (HsTOP1) is composed of four domains: one variable and putatively unstructured N-terminal domain (residues 1–213), a conserved core domain (residues 214–635), a flexible linker domain with variable length (residues 636–712), and one highly conserved C-terminal domain (residues 713–765) containing the catalytic tyrosine residue (Fig. 1a). Several TOP1 structures have been solved, from either human or *Leishmania donovani* TOP1, in both covalent and noncovalent complexes with DNA[10,12–21]. In these structures, TOP1 forms a toroidal fold, in which two modules entrap the DNA molecule (Fig. 1b). The capping module (residues 214–426) corresponds to the first half of the core domain also named the CAP domain, while the catalytic module (residues 435–765) comprises the second half of the core domain also named the CAT domain (residues 435–635), the linker domain (residues 636–712) and the C-terminal domain (residues 713–765) (Fig. 1b). In particular, two loops from the CAP and CAT domains called the "Lips", form a salt bridge, stabilizing this closed conformation[18,22].

In all the available structures of eukaryotic TOP1 enzymes, only one conformational state (closed conformation) has been solved so far[10,12–21]. At the beginning of our work, there was no information about how the enzyme opens and closes to enable DNA entry and exit. Several groups have studied non-eukaryotic TOP1 orthologs, including vaccinia and *Deinococcus radiodurans* Topo IB to obtain structural information[23,24]. Such approaches have provided insights into the catalytic region (CAT and C-terminal domains), which is conserved between bacterial, viral and eukaryotic Topo IB (Fig. 1)[25–27]. Nonetheless, bacterial and

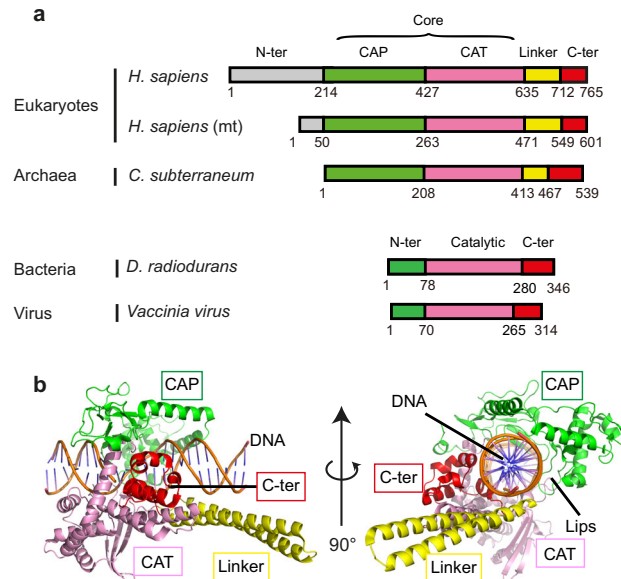

**Fig. 1 Domains composition of type IB topoisomerases. a** Schematic view of the domain organization of eukaryotic, archaeal, bacterial, and viral Topo IB. The CAP, CAT, linker, and C-terminal domains are shown as green, pink, yellow, and red boxes, respectively. The N-terminal domains are shown in gray for Eukarya and Archaea Topo IB enzymes and green for Bacteria and Virus for Topo IB enzymes. **b** Cartoon representation of the Human TOP1 (HsTOP1) crystal structure (PDB 1K4T). Coloring is the same as in **a**.

viral Topo IB possess a different capping module so that the internal motion of the enzyme is likely different compared to eukaryotic TOP1.

In 2008, topoisomerase IB was discovered in several archaea that form a superphylum, the Baty-Aig-Thaumarchaeota (BAT)[6,28]. In particular, one topoisomerase from the hyperthermophilic archaeon *Caldiarchaeum subterraneum* (CsTOP1) was found to be highly similar to human TOP1 (34% sequence identity, 53% similarity) and to display similar activity, while being extremely stable[28]. In this work, we solved the CsTOP1 crystal structure at 2.0 Å resolution. This structure reveals how archaeal topoisomerases IB fold, and provides key information regarding TOP1 conformation in the absence of DNA. Similar to what was observed with bacterial Topo IB, in the absence of DNA, CsTOP1 is folded in an open form. Our structure and biochemical analyses reveal how one region between the CAP and the CAT domains, the hinge, is responsible for the opening/closing of the TOP1–DNA cavity, while the "Lips" on the opposite side of the protein, form a salt bridge to lock the closed conformation. Altogether, our work brings structural details about the motion of TOP1 enabling the entry/exit of the DNA substrate, which is reversibly cleaved by TOP1.

## Results

**Crystal structure of CsTOP1 apo-form.** The eukaryotic TOP1 structure has been solved with a DNA molecule, in a closed conformation (Fig. 1b). To understand how TOP1 folds in the absence of its substrate, we produced crystals of full-length CsTOP1 in the absence of DNA. We solved the crystal structure of both selenomethionine-labelled (SeMet) CsTOP1 and native CsTOP1 at 2.0 and 2.20 Å, respectively. In both cases, the asymmetric unit of the crystal contained two TOP1 molecules (Supplementary Fig. 1). In the asymmetric unit, each TOP1 molecule folds similarly, and there is no significant difference between SeMet CsTOP1 and native CsTOP1 (Supplementary

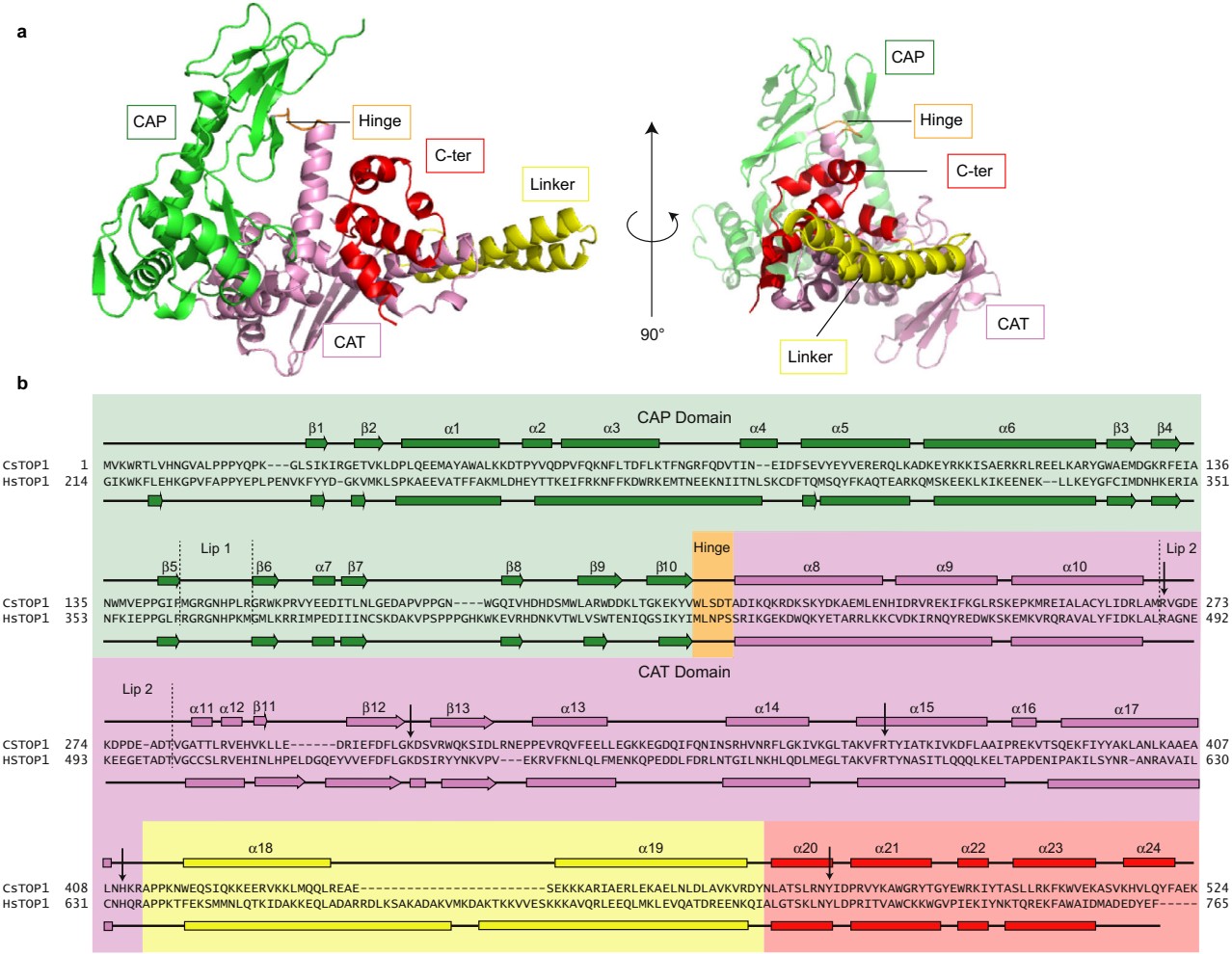

**Fig. 2 Crystal structure of archaeal CsTOP1. a** Open conformation of CsTOP1 solved by crystallography. **b** Sequence alignment between CsTOP1 and HsTOP1. The position of the lips are indicated with dotted lines. The residues interacting with DNA are shown with arrows. The coloring scheme is the same as in Fig. 1, apart from the Hinge loop which is shown in orange.

Fig. 1). In its apo-form, CsTOP1 folds as a compact protein, comprised of the CAP region (1–207), the CAT region (208–412), the linker (413–469), and the C-terminal region (469–524) (Fig. 2a). The electron density of the C-terminal extremity of CsTOP1 (525–539) is not visible in these crystal structures, suggesting that this region, which is poorly conserved in other archaea, is flexible (Supplementary Fig. 2). Notably, the CsTOP1 structure is largely different from the previously solved eukaryotic TOP1 structures (Fig. 1b). In particular, the CAP and CAT domains do not form a toroidal structure with the cavity for DNA binding (Fig. 2a), suggesting that TOP1 binding to DNA is associated with a large reshaping of the protein with enclosure of the DNA inside the enzyme.

Despite the overall difference between CsTOP1 and HsTOP1-DNA structures, these enzymes share very similar sequences and secondary structures (Fig. 2b). In particular, the key residues involved in DNA binding, the two Lips, and the catalytic tyrosine align perfectly. The main difference is the shortening of the linker α-helices in CsTOP1 and in most archaeal TOP1. Interestingly, several Asgard archaea genomes encode a TOP1 with a substantially longer linker, which may suggest some difference in the enzymatic activity (Supplementary Fig. 2).

**Entry of a DNA molecule is associated with the rotation of the CAP region**. Taken separately, the individual domains of the HsTOP1 closed conformation and of the CsTOP1 open-conformation superpose relatively well. In particular, the CAP and the C-terminal domains superpose almost completely with an RMSD of 1.1 Å for 171 Cα atoms and 1.1 Å for 43 Cα atoms, respectively; thereby supporting the high similarity of the two enzymes (Fig. 3a). The CAT domains also superpose well with an RMSD of 2.4 Å for 177 Cα atoms with some local differences (Fig. 3a). These differences include the second "Lip" loop (Lip2) that is closer to the DNA cavity in the open-conformation contrary to the first "Lip" loop (Lip1) from the CAP domain, which remains at the same position in both the open and closed conformations. Apart from this, the overall difference between the open and the closed conformations is caused by different spatial arrangements of the different TOP1 domains. This suggests that these domains can move relatively to each other during DNA entry/exit. More precisely, upon DNA binding, the CAP domain appears to rotate in order to cap the DNA molecule and form the toroidal fold with the CAT domain (Fig. 3b). The interaction between the two Lips, which is essential for TOP1 activity[29], can only occur in the closed conformation (Fig. 3b), suggesting that

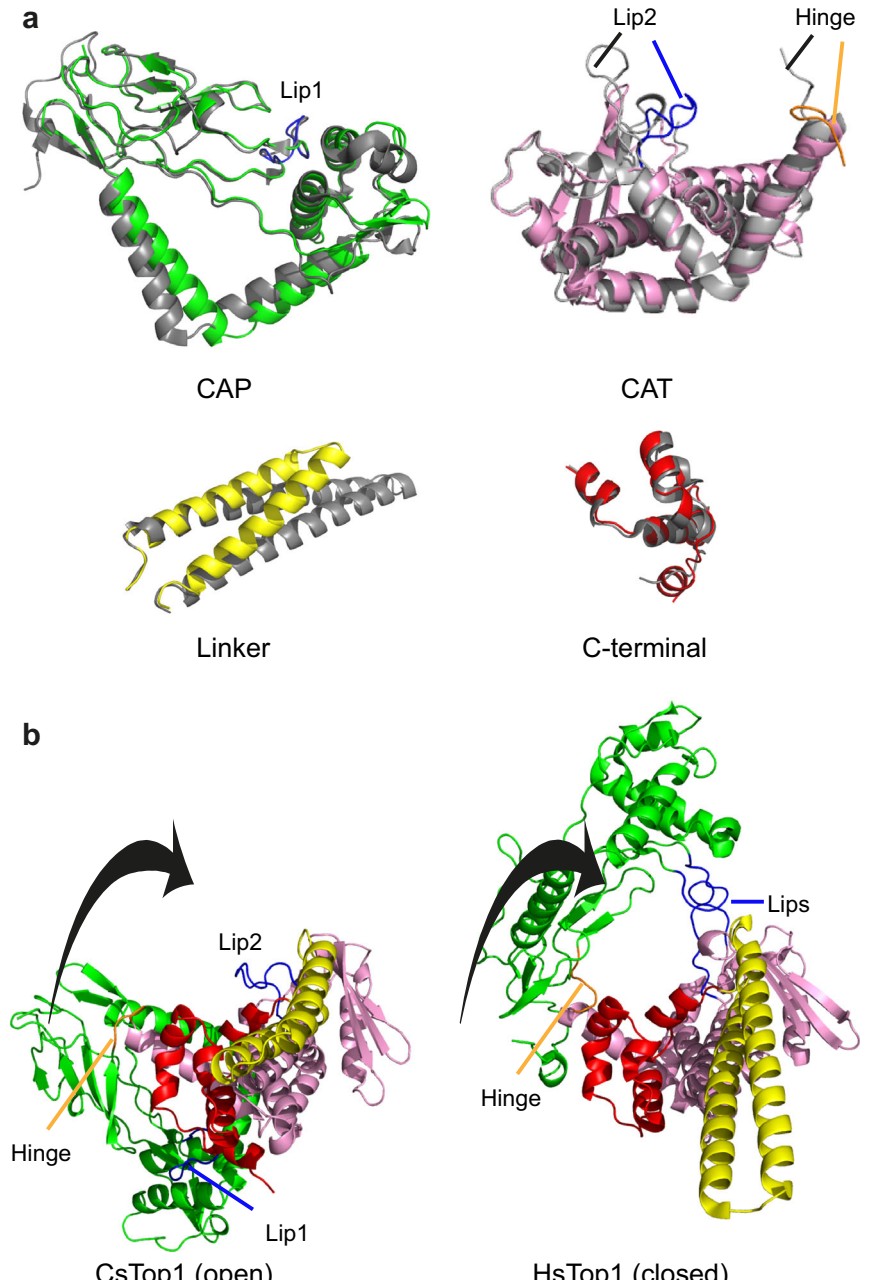

**Fig. 3 Comparison of the TOP1 domains in their open and closed conformations. a** Superimposition of the CAP, CAT, Linker, and C-terminal domains of CsTOP1 and HsTOP1. CsTOP1 coloring is the same as in Fig. 1 and HsTOP1 is shown in gray. **b** Rotation of the CAP domain during DNA binding. The coloring is the same as in Fig. 1. The black arrow represents the rotation of the CAP domain from the open to the closed conformation.

the salt-bridge formation "locks" TOP1 in this state after rotation of the CAP domain.

**The DNA-binding interface of TOP1 is partially accessible in its open conformation.** Based on the analogy with HsTOP1[30], CsTOP1-DNA interactions involve mainly five residues including the catalytic tyrosine (Tyr477) in the C-terminal domain and four conserved residues named the catalytic tetrad (Arg269, Lys306, Arg367, and His410). These five residues are not only conserved between human and archaeal TOP1 (Fig. 2b), but their relative positions are also similar between the open and closed enzyme conformations (Fig. 4a). This is in contrast with poxvirus Topo IB, in which the DNA interacting regions are partially unfolded in the apo-form and the DNA molecule is required for the

repositioning of the catalytic tetrads[31]. In archaeal TOP1, entry of the DNA molecule has little effect on this central DNA interacting region, which is folded prior to DNA binding.

In the apo-form structure, the TOP1 CAP domain does not block access of the DNA molecule to the DNA interacting region (Fig. 3b). The Lip2 loop segment moves towards this region and partially covers it (Fig. 4a). More precisely, the side chain of glutamate Glu278 (Lip2) forms hydrogen bonds with the Nε atom of His410 and the hydroxyl group of Tyr477 (Fig. 4b). In addition, the Arg269 side chain forms hydrogen bonds with the Lip2 Thr281 main chain and Glu273 side chain. Therefore, while the DNA cavity is not blocked by the presence of the CAP domain in the CsTOP1 open-conformation, the Lip2 loop interaction with the catalytic tetrad may hinder the entry of the DNA molecule, and its displacement seems necessary for DNA binding.

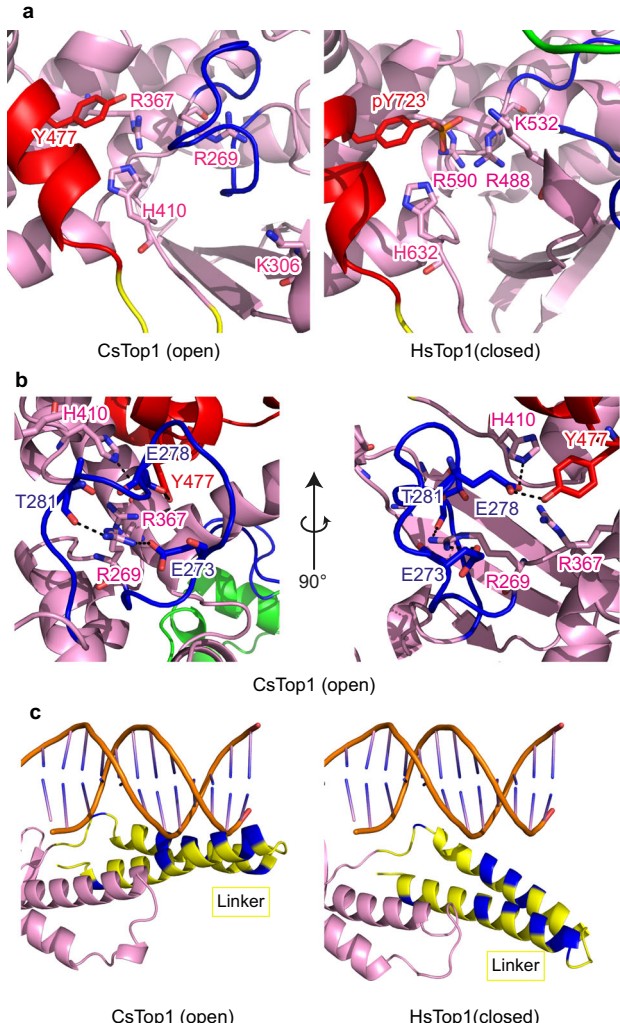

**Fig. 4 DNA-binding region of TOP1 in the open and closed conformations.** **a** Comparison of the relative positions of DNA interacting residues of CsTOP1 and HsTOP1. **b** Interaction between the lip2 and DNA interacting residues in the open state CsTOP1. CsTOP1 coloring is the same as in Fig. 1. **c** Position of the linker and DNA in the HsTOP1 crystal structure (PDB 1K4T)[10] (right panel). For the left panel, the putative position of the DNA molecule interacting with CsTop1 is calculated after superposition using Pymol of CsTop1 structure with HsTOP1-DNA structure (PDB 1K4T). The basic residues are colored blue.

**Orientation of the linker domain.** In addition to the five canonical residues involved in DNA binding, the linker domain, which is rich in lysines and arginines, has also been reported to interact with DNA[30]. Surprisingly, in all HsTOP1-DNA structures, this domain is orientated 30° away from the DNA duplex axis, so that the lysine- and arginine-rich tip of the linker does not interact with DNA (Fig. 4c)[12,13]. Notably, the orientation of this linker domain is different in the open-form crystal structure, with this domain becoming almost parallel to the DNA cavity axis (Fig. 4c). In this conformation, the DNA molecule can interact with the catalytic site of TOP1 and with the basic residues (arginine and lysine, in blue) of the linker domain (Fig. 4c). It is surprising that in all HsTOP1-DNA structures, the linker was consistently bent away from the DNA molecule[10,12–21]. This could be due to crystal packing, the linker domain being pushed away from the DNA axis by the CAT domain from another TOP1 molecule (Supplementary Fig. 3). Thus, crystal constraints may explain the odd position of the linker domain in the HsTOP1-

DNA crystal structures. Our CsTOP1 structural data highlight the possibility for the linker domain to remain close to the DNA molecule in the different steps of the topoisomerase reactions.

**TOP1 internal motion relies on a five-residue loop, the hinge.** On the other side of the protein, another small protein loop shows different orientations in the closed and open conformations. This loop, the hinge, is found at the junction between the CAP and the CAT domains (residues 209–213) (Fig. 2b). In the closed conformation, the hinge is pointing towards the top of the protein, so that the CAP domain is located above the DNA molecule, closing the DNA cavity (Fig. 3b). In the open conformation, the hinge is orientated towards the side of the protein, opening the DNA cavity (Fig. 3b). Hence, we propose that the large internal rotation of TOP1 appears to rely on this five-residue loop.

Alignment of the different TOP1 sequences shows that both eukaryotic and archaeal enzymes possess this five-residue loop (Fig. 5a). Among these residues, one leucine (Leu210 in CsTOP1) is conserved in archaea and eukaryotes. In both the HsTOP1-DNA and CsTOP1 structures, the side chain of this residue fits in a conserved hydrophobic pocket from the CAP domain (Fig. 5b). Therefore, this leucine appears to be important for coupling the motions of the loop with the full CAP domain. In the vicinity of this leucine, a tyrosine (Tyr207 in CsTOP1) is highly conserved in both archaeal and eukaryotic TOP1s (Fig. 5a). This residue is also conserved in vaccinia Topo IB, highlighting its importance[32]. In the closed conformation, this tyrosine (Tyr426 in HsTOP1) is one of the few residues from the CAP domain interacting with the DNA duplex, while in the open conformation it is pointing away from the DNA cavity (Fig. 5c). Hence, this tyrosine may be important to guide the motion of the CAP domain upon DNA binding, enabling the enzyme closing. Accordingly, Y207A mutation abrogated the activity of the enzyme in vitro (Supplementary Fig. 4a).

By contrast with these two highly conserved residues (Leu210 and Tyr207), the last three residues of the hinge are less conserved. In particular, in the eukaryotic enzymes, this part of the hinge comprises high flexibility amino acids (GAG in *A. thaliana*) or more rigid residues (NPS in *H. sapiens*) (Fig. 5a)[33]. These sequence differences may be related to differences in the structural flexibility of the hinge. In the archaeal enzymes, these three residues follow the pattern XDS/T, suggesting that the hinge sequence can accommodate little variation in this domain of life (Fig. 5a). To confirm this observation, we replaced the archaeal hinge domain (WLSDT) with the human sequence (MLNPS) and checked the ability of the enzyme to relax supercoiled DNA (Fig. 5d). While purified CsTOP1 relaxed DNA in 30 s, the same amount of mutated CsTOP1 could only partially relax DNA even after 6 min. We observed a similar impediment when replacing hinge residues with proline (WLSPP). This demonstrates the importance of the flexible hinge loop for efficient TOP1 activity. Replacement of hinge residues with glycine (WLSGG) also decreases TOP1 activity (Supplementary Fig. 4a), highlighting that there is no direct correlation between rigidity and activity of the enzyme. The divergent hinge sequences in eukaryotic organisms suggest that TOP1 enzymes have evolved different hinge loops to produce different dynamics of opening/closing of the active site.

**TOP1 sensitivity to camptothecin.** Unlike human TOP1, CsTOP1 is resistant to the antitumoral drug camptothecin (CPT)[28], which reversibly traps the eukaryotic TOP1–DNA covalent complex (TOP1cc) by slowing down the religation of the cleaved DNA strand[34]. Even though CsTOP1 is unaffected by

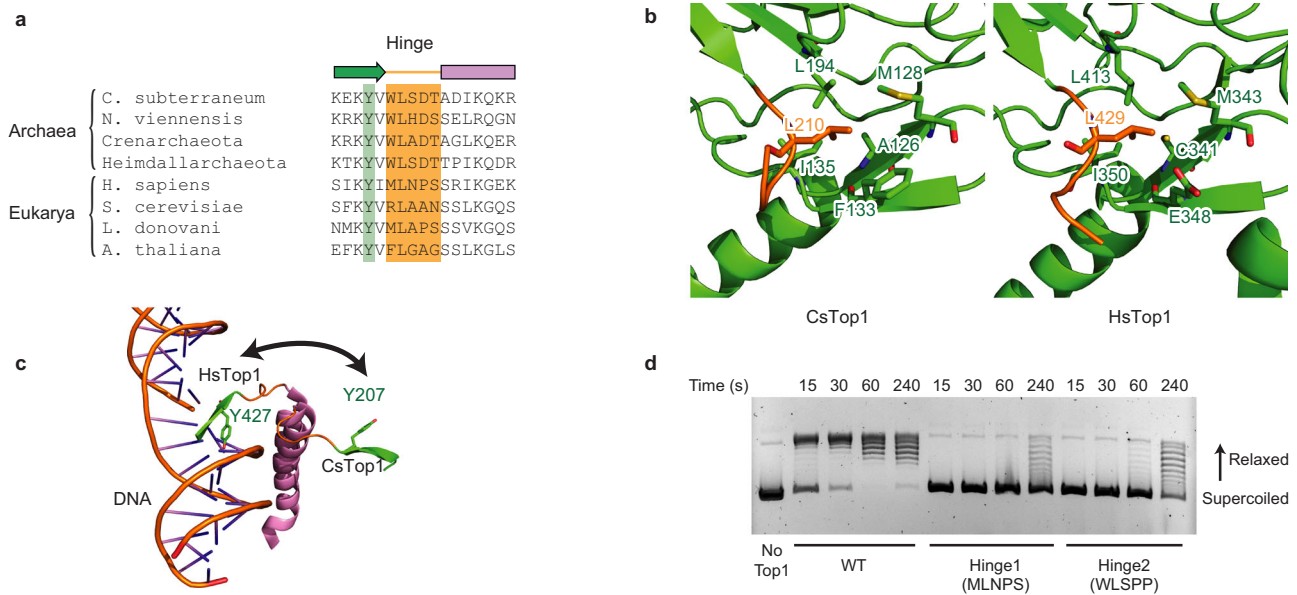

**Fig. 5 TOP1 hinge loop. a** Alignment of several archaeal and eukaryotic hinge loops, highlighted in orange. The conserved tyrosine is highlighted in green. **b** Superimposition of the HsTOP1 and CsTop1 CAP domains. The hydrophobic interaction between the conserved leucine from the loop and one conserved hydrophobic pocket in the CAP domain is shown. The coloring scheme is the same as in Fig. 1. **c** Structural superimposition of the open and closed conformations of CsTOP1. The motion of the hinge allowing the interaction between one conserved tyrosine and DNA is shown. **d** Relaxation of negatively supercoiled DNA by WT and hinge mutated CsTOP1. The kinetics of relaxation is shown for the three constructs. The experiment was done twice independently.

CPT, it forms spontaneously stable TOP1 cleavage complexes independently of CPT[28]. We thus used the structural information from both HsTOP1 and CsTOP1 enzymes to understand the cause of this difference, which could result from a local difference in the protein, preventing the physical interaction between TOP1 and CPT, or from a difference in the enzyme dynamics. Indeed, HsTOP1 sensitivity to CPT has been related to the dynamics of TOP1 activity. The DNA rotation rates determine a short window of time during which the DNA is cleaved by TOP1, and the nicked DNA forms the pocket that accommodates the camptothecin drugs[35–37].

To elucidate how CPT affects different mutants of CsTOP1, we developed a DNA cleavage assay to distinguish between CPT-induced and CPT-independent DNA cleavage. It is known that CPT traps TOP1cc at preferential positions[38], which differ from the CsTOP1cc preferential positions[28]. We designed a 30 bp oligomer containing both preferential DNA sequences (Fig. 6a). Incubation of CsTOP1 with this DNA generated 8- and 11-nucleotide DNA cleavage fragments (Fig. 6b). These TOP1ccs were similar in the presence and absence of CPT and were reversed by salt addition. By contrast, incubation of HsTOP1 induced TOP1ccs exclusively in the presence of CPT, and those TOP1ccs were relatively resistant to salt reversal (Fig. 6b).

**TOP1 dynamics and camptothecin sensitivity.** Thereafter, we used the cleavage assay with CsTOP1 mutants to determine the link between TOP1 dynamics and CPT-dependent TOP1ccs. Mutations were based on known positions involved in TOP1 dynamics (Table 1). Those include the catalytic-dead TOP1 (CsTOP1^{Y477F}), which had been shown to have no topoisomerase activity[28]. As expected, CsTOP1^{Y477F} did not cleave DNA (Fig. 6c). We also replaced in CsTOP1 the residues surrounding the catalytic tyrosine (residues LRNYI) by the corresponding human residues (residues KLNYL). This triple mutation did not affect the overall activity of the enzyme, as shown in DNA relaxation assays (Supplementary Fig. 4b). CsTOP1^{HsTyr}

displayed a DNA cleavage pattern similar to WT CsTop1 with only CPT-independent TOP1ccs (Fig. 6c). These results suggest that the difference between the archaea and human TOP1s cannot be explained by the local difference between the two enzymes around the cleavage site. We also fused part of the human N-terminal domain (residues 191–215) to the N-terminal extremity of CsTOP1, which is known to stabilize the closed conformation in human TOP1 and to promote CPT sensitivity[39]. This CsTOP1^{HsNter} mutant did not produce CPT-dependent TOP1cc (Fig. 6c).

By contrast, replacing the linker and the C-terminal domains with human sequences (CsTOP1^{HsLinker-Cter}) led to the formation of the 24-nt DNA fragment after CPT treatment (Fig. 6c). Hence, the difference between CsTOP1 and HsTOP1 regarding both their DNA nicking-closing activity (C-terminal domain) and relaxation (linker domain) activities determines CPT sensitivity. In HsTOP1, salt bridges between the linker and one α-helix from the CAT domain (helix α17 in Fig. 2) appear to be involved in enzyme flexibility and CPT sensitivity. These bridges are absent in the TOP1 enzymes from CPT-producing plants[40] and mutations abolishing these salt bridges have been identified in CPT-resistant human cell lines[41]. In the chimera CsTOP1^{Hslinker-Cter-L399R L402R}, we restored these salt bridges between helix α17 and the human linker. The resulting protein showed stronger CPT-dependent DNA cleavage (24-nt DNA) compared to CsTOP1^{Hslinker-Cter} (Fig. 6c). Thus, the flexibility of the linker domain appears an important factor for CPT sensitivity, in accordance with independent studies[35,36,42,43].

We next tested whether stabilization of the closed conformation could be involved in CPT sensitivity in our chimera. Indeed, interactions between the two Lips of HsTOP1 have been suggested to be important for CPT sensitivity, and mutating the Lip1 lysine 369 has been shown to decrease CPT sensitivity in human cells[29]. This residue being replaced by the leucine 154, we constructed the mutation L154K in the CsTOP1^{Hslinker-Cter} context. This mutation enhanced CPT-dependent TOP1cc with

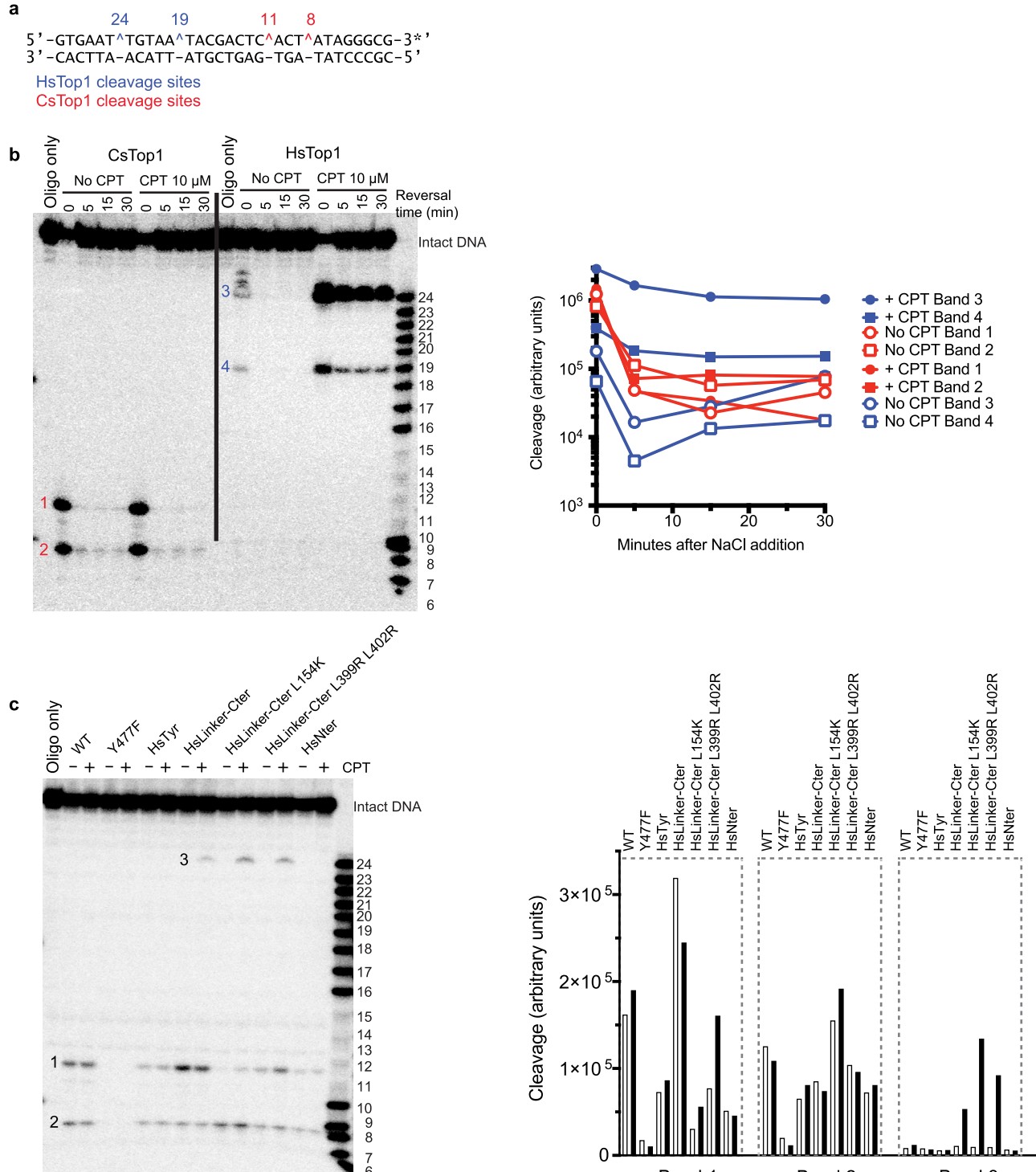

**Fig. 6 TOP1-mediated DNA cleavage assays. a** Sequence of the oligonucleotide used in the assays. The HsTOP1 cleavage site (CPT-dependent) and CsTOP1 cleavage site (CPT-independent) are shown in blue and red, respectively. The star represents the position of $^{32}$P labelling. Ladder numbering represents the size of the DNA in nucleotides. **b** Cleavage of the oligonucleotide with WT CsTOP1 and HsTOP1 in the presence or absence of CPT. The reversion of TOP1ccs is monitored after the addition of 350 mM NaCl. **c** Cleavage of the oligonucleotide by WT and the indicated biologically engineered CsTOP1 (chimera) in the presence or absence of CPT. This experiment was done twice.

little impact on the CPT-independent TOP1ccs (Fig. 6c). This result suggests that CsTOP1 displays a weaker Lips-related stabilization of the closed conformation compared to HsTOP1, and that this difference contributes to the resistance of CsTOP1 to CPT.

Altogether, our data suggest that, while the hinge seems important to enable the transition from the closed to the open conformation, the stabilization of the closed state relies on different regions of TOP1, notably the linker domain and the Lips. Our hypothesis is that these two domains impact the

**Table 1 CsTOP1 constructs used for CPT sensitivity assays.**

| Construct name | Description | Predicted effect | Cleavage pattern |
|---|---|---|---|
| HsTop1 | Wild-type human enzyme | Camptothecin sensitivity | CPT-dependent cleavage bands |
| CsTop1 | Wild-type archaeal enzyme | Camptothecin resistance | CPT-independent cleavage bands |
| CsTOP1[Y477F] | Catalytic-dead enzyme | No cleavage of DNA | No cleavage |
| CsTOP1[HsTyr] | Residues surrounding the catalytic tyrosine are replaced by human residues | DNA cleavage similar to HsTOP1 | Similar to CsTop1 |
| CsTOP1[HsNter] | A human N-terminal extension was added to CsTOP1 | Stabilization of the closed conformation | Similar to CsTop1 |
| CsTOP1[HsLinker-Cter] | Both linker and C-terminal domains are replaced by human domains | DNA cleavage and relaxation similar to HsTOP1 | CPT-independent cleavage bands and low amount of CPT-dependent cleavage bands |
| CsTOP1[Hslinker-Cter-L154K] | CsTOP1[Hslinker-Cter] in which Lip1 loop can interact strongly with Lip2 loop | DNA cleavage and relaxation similar to HsTOP1 + stabilization of the closed conformation | CPT-independent cleavage bands and a moderate amount of CPT-dependent cleavage bands |
| CsTOP1[Hslinker-Cter-L399R L402R] | CsTOP1[Hslinker-Cter] in which the linker makes salt bridges with the CAT domain | DNA cleavage and relaxation similar to HsTOP1 + reduced flexibility of the linker domain | CPT-independent cleavage bands and a moderate amount of CPT-dependent cleavage bands |

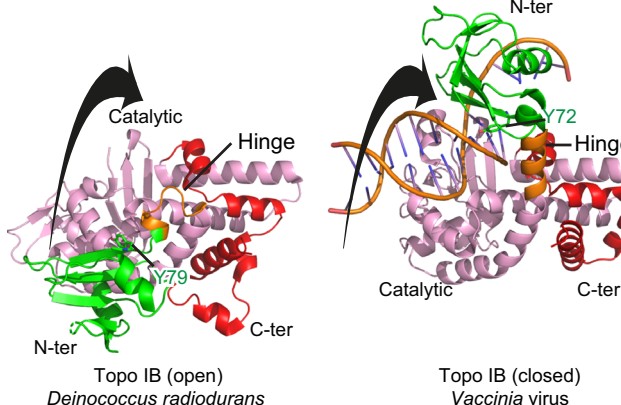

**Fig. 7 Comparison of short Topo IB domains in their open and closed conformations.** Superimposition of Topo IB from *Deinococcus radiodurans* (PDB 2F4Q) and from Vaccinia (PDB 2H7G). Coloring is the same as in Fig. 1. The black arrow represents the rotation of the CAP domain from the open to the closed conformation.

geometry and/or the kinetics of the opening/closing of the active site, and determine the stacking of CPT inside the pocket formed by the base pairs flanking the nicked DNA. Further molecular work based on single-molecule assays is warranted to dissect this CPT sensitivity/resistance mechanism and the controlled rotation of the nicked DNA within CsTOP1ccs.

**Internal motion in the small type IB topoisomerases.** The type IB topoisomerase family comprises short enzymes encoded by bacteria and viruses. Notably the topoisomerase IB from *variola* virus has been extensively studied[23] (Fig. 1a) and several structures of the *variola* virus TOP1–DNA complex have been solved showing similarities between the human and viral Topo IB catalytic domains (Fig. 7)[27,44]. The structure of the bacterial *Deinococcus radiodurans* Topo IB has been solved as well in the absence of DNA[25,26], and like our structure, the comparison between the apo-form and the DNA-bound topoisomerase structures suggest rotation of the N-ter domain after DNA binding (Fig. 7). Like the archaeal and eukaryotic TOP1 enzymes,

the bacterial and viral enzymes contain a tyrosine (Y79 and Y72, respectively) that interacts with DNA in the closed conformation. Thus, the mechanism involved in the opening/closing of Type IB topoisomerases appears conserved from virus/bacteria to eukarya/archaea.

## Discussion

Based on our structural and functional data, we propose a mechanistic model enabling the transition from the open to the closed conformation of TOP1 enzymes (Fig. 8). In its apo-form, the TOP1 CAP domain rotates around the hinge while the DNA-binding region interacts with Lip2. After the displacement of this loop, the DNA molecule can enter the DNA cavity and promote the motion of the CAP domain via a conserved tyrosine (Tyr207 in CsTOP1 and Tyr426 in HsTOP1). This closes TOP1 in a toroidal fold entrapping the DNA. Interaction between the two Lips can then stabilize this closed conformation, during the DNA cleavage and relaxation steps.

In their seminal TOP1 structure paper, Champoux and colleagues proposed that a hinge region was necessary to open and close the enzyme[13]. Based on limited proteolysis results, they predicted one flexible region around Pro431, similar to the hinge region (428–432) we identified in our present structural work. In 2009, Desideri and colleagues took advantage of the bacterial *Deinococcus radiodurans* Topo IB structure (Fig. 1a) to go further into modeling the enzyme opening. *D. radiodurans* Topo IB was at that time the only member of the Topo IB family whose structure had been solved without DNA[25]. They concluded that the TOP1 catalytic site opening relied on a hinge region that corresponds to the destabilization of the α-helix 8 (434–465), which is less rigid between residues 437–444[24]. Their prediction is consistent with our results, as we identified a hinge corresponding to the loop right before helix 8 (residues 428–432). Nonetheless, we did not observe the helix destabilization, which may be specific to bacterial and viral Type IB topoisomerases that are quite distant from eukaryotic and archaeal TOP1[6]. One cannot exclude that this destabilization occurs during DNA supercoil relaxation. Indeed, previous molecular dynamics simulation suggested that the CAP and CAT modules move during relaxation with a large stretching of the hinge region[45]. However, the hinge is relatively small (five residues) with one leucine physically constrained by its anchored position, and one proline residue constrained by its inherent chemical property. This stretching seems physically

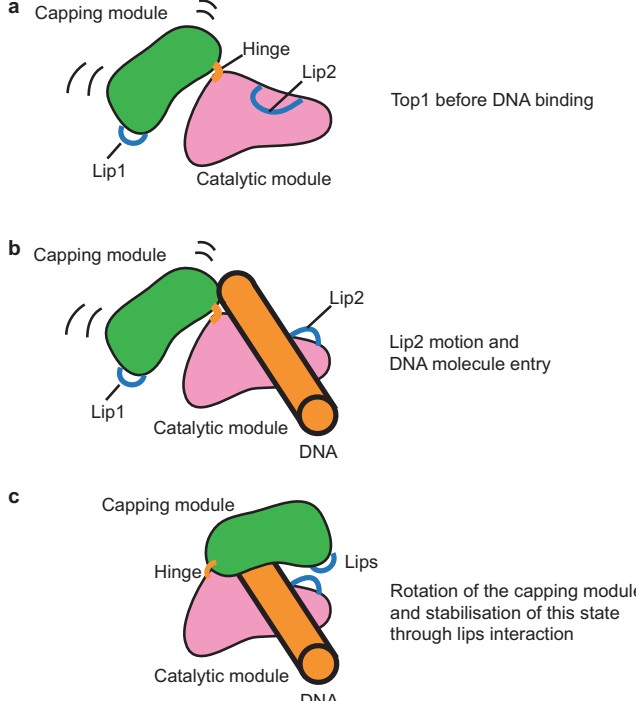

**Fig. 8 Proposed model of DNA entry inside the TOP1 catalytic site. a** In its apo-state, the CAP domain (green) and the CAT, Linker, C-ter domains (pink) can rotate. Lip2 is interacting with the DNA-binding region of the catalytic module. **b** After the displacement of the Lip2, one DNA molecule can enter the TOP1 catalytic site. **c** The Capping module rotates to trap DNA in the catalytic site. This rotation relies on the hinge and one conserved tyrosine from the CAP domain. The interaction between the two lips stabilizes the closed conformation.

impossible unless it involves rearrangement of other regions, such as the following α-helix destabilization. Hence, the role of the hinge in supercoils relaxation warrants further studies.

The differences in enzyme dynamics between bacterial/viral Topo IB and eukaryotic/archaeal TOP1 enzymes suggest that one should be careful when extrapolating observations from one group of Topo IB enzymes to the other. Still, CsTOP1 and HsTOP1 structural similarities imply that some features of the archaeal enzyme can be extrapolated to eukaryotic TOP1 enzymes.

The extreme thermostability of CsTOP1 is a unique property that can be exploited for further studies and more specifically for biophysical assays. In particular, CsTOP1 could be used further to solve other states of the enzyme during its topoisomerase reactions, or to solve the structure of CsTOP1 with particular substrates, such as G-quadruplexes[46].

One potential question arising from our model (Fig. 8) is that it involves a single DNA molecule while it has been proposed that TOP1 can recognize DNA crossings in vitro and could simultaneously bind two DNA duplexes both in the case of the human and viral enzymes[47,48]. There is currently little information on a secondary DNA-binding region, apart from one structure with bacterial Topo IB[26]. It would be informative to determine such structures with eukaryotic/archaeal TOP1 and elucidate whether this secondary DNA binding affects the entry of the substrate DNA molecule cleaved by TOP1. One could speculate that this secondary DNA binding could destabilize the interaction between Lip2 and the DNA-binding region, facilitating access of the substrate DNA to the TOP1 active site.

## Methods

**TOP1 constructs**. CsTOP and CsTOP1[Y477F] expression plasmids have been constructed in another study[28]. Briefly, the gene encoding the *Caldiarchaeum subterraneum* [GenBank BAJ49459.1] CsTOP1 was codon-optimized and cloned into the pETM11 vector using *NcoI* and *XhoI*. The resulting construct encoded CsTOP1 with a Hexa-histidine tag and a TEV cleavage site. The Chimera construct CsTOP1[Hslinker-Cter] corresponds to CsTOP1 in which residues 411–551 are replaced by residues 632–765 from HsTOP1. This construct was codon-optimized and cloned into pETM11 similarly to the wild-type (WT) enzyme. Codon-optimised sequences are shown in Supplementary Table 1. Different mutations were introduced using the Q5® Site-Directed Mutagenesis Kit (NEB). They include L154K for CsTOP1[Hslinker-Cter-L154K], L399R, L402R for CsTOP1[Hslinker-Cter-L399R L402R], Y207A mutations for CsTop1[Y207A] and the Hinge 1, Hinge 2, and Hinge 3 mutations (respectively, corresponding to the sequences MLNPS, WLSPP, and WLSGG). We also used this approach to produce the chimera construct CsTOP1[HsTyr] and CsTOP1[HsNter]. In the case of CsTOP1[HsTyr], the residues surrounding CsTOP1 catalytic site (KLNYL) were replaced by HsTOP1 sequence (LRNYI). Chimera construct CsTOP1[HsNter] corresponds to CsTOP1, in which residues 191–213 from HsTOP1 were fused to its N-terminal extremity. Primers are shown in Supplementary Table 2.

**Sample preparation**. *Escherichia coli* strain BL21(DE3) cells were transformed with each expression vector and cultured in LB medium containing 50 mg L$^{-1}$ kanamycin at 37 °C. When the optical density of the culture at 600 nm reached ~0.5, protein expression was induced by the addition of isopropyl-β-D-thiogalactopyranoside (IPTG) to a final concentration of 1 mM. The culture was continued for 1–2 h at 37 °C. Cells were suspended in a 50 mM Tris-HCl (pH 8.0), 2 M NaCl and 20 mM imidazole buffer containing a protease inhibitor cocktail (Sigma–Aldrich) and lysed by sonication. In the case of CsTOP1, the lysed cells were further incubated at 75 °C for 15 min. Chimera constructs were not incubated at high temperature and were directly centrifuged for 60 min at 15,000 rpm at 4 °C. The supernatant was filtered through a 0.2 μm filter membrane and loaded onto a 5 mL Ni-NTA column (HisTrap HP, GE Healthcare). Samples were eluted with a linear gradient of 0–1 M imidazole. The peak fractions were collected, concentrated, and subjected to a 1 mL heparin affinity column (HiTrap Heparin HP, GE Healthcare) in 50 mM Tris-HCl (pH 8.0) buffer containing 200 mM NaCl. Samples were eluted with a linear gradient of 0.2–2 M NaCl, and fractions containing the purified protein were collected, concentrated and stored at −80 °C until use.

The selenomethionine (SeMet)–labeled mutant was produced in BL21(DE3) *E. coli* cells as follows. Cells were grown in M9 medium at 37 °C until they reached an optical density of 0.63. Thereafter lysine (100 mg/L), threonine (100 mg/L), phenylalanine (100 mg/L), leucine (50 mg/L), isoleucine (50 mg/L), valine (50 mg/L), and selenomethionine (50 mg/L) were added to the medium, and cells were grown for an additional 30 min before induction with 1 mM of IPTG for 2 h at 37 °C. After cells harvesting, SeMet CsTop1 was purified as above.

**Crystallization and data collection**. Native CsTOP1 was prepared at 11 mg/ml in 200 mM KCl, 40 mM Tris-HCl (pH 8.0). Crystallization experiments were performed by the sitting drop vapor diffusion technique in 96-well plates, according to established protocols at the Crystallography Core Facility of the Institut Pasteur[49].

The best crystals were manually reproduced at 20 °C with the hanging drop vapor diffusion method by mixing 1 μL of the protein solution with an equal amount of precipitant solution containing 22% PEG 6000, 1 M LiCl 100 mM sodium acetate. For data collection, crystals were flash-cooled in liquid nitrogen using a paratone/paraffin oils mixture (50%/50%) as a cryoprotectant. SeMet CsTOP1 was prepared at 6 mg/ml in 200 mM KCl, 40 mM Tris-HCl (pH 8.0). Crystals were grown at 20 °C with the sitting drop vapor diffusion method by mixing 1 μL of protein solution with an equal amount of precipitant solution containing 24% PEG 6000, 1 M LiCl 100 mM sodium acetate. For data collection, the SeMet CsTOP1 crystals were flash-cooled in liquid nitrogen using a paratone/paraffin oils mixture (50%/50%) as a cryoprotectant.

The X-ray diffraction data were collected on beamline PROXIMA-2 (Synchrotron SOLEIL, St Aubin, France) or beamline ID-30 (Synchrotron ESRF, Grenoble, France). The diffraction images were integrated with the program XDS[50] and crystallographic calculations were carried out with programs from the CCP4 program suite[51]. The selenium sites were located using SHELXD[52]. Single-wavelength anomalous diffraction (SAD) phasing and density modification were carried out with the program AutoSHARP[53]. The resulting electron density map allowed chain tracing for most of the polypeptide chain using the program Coot[54]. Crystallographic refinement was carried out with Phenix[55] alternated with rounds of validation and rebuilding with Coot[54]. The experimental (native and selenomethionine) structure factors and the refined coordinates for the final model (Table 2) have been deposited in the Protein Data Bank with accession codes 6Z01 and 6Z03, respectively.

**DNA relaxation assay**. Relaxation assays were performed in a final volume of 20 μl with 0.33 μg negatively supercoiled pBR322 plasmid and 0.16 μg enzyme per reaction. The reaction buffer contained 10 mM Tris-HCl (pH 8.5), 0.1 mM EDTA,

**Table 2 Data collection and refinement statistics.**

| | CsTOP1-SeMET | CsTOP1-native |
|---|---|---|
| Wavelength (Å) | 0.979030 | 0.966000 |
| Resolution range (Å) | 34.98–2.00 | 70.07–2.20 |
| | (2.07–2.00) | (2.24–2.20) |
| Space group | P 1 21 1 | P 1 21 1 |
| Unit cell | | |
| *a, b, c* (Å) | 76.2 96.6 94.6 | 75.5 94.7 92.7 |
| *α, β, γ* (°) | 90.0 113.3 90.0 | 90.0 111.8 90.0 |
| Unique reflections | 84,809 (8470) | 61,691 (3039) |
| Multiplicity | 9.2 (9.5) | 6.8 (7.2) |
| Completeness (%) | 99.8 (100) | 99.9 (100) |
| *I/σI* | 19.2 (4.3) | 14.3 (2.2) |
| Wilson *B*-factors | 30.53 | 42.36 |
| *R*-merge | 0.084 (0.459) | 0.085 (0.982) |
| *R*-meas | 0.090 (0.486) | 0.092 (1.086) |
| *R*-pim | 0.030 (0.157) | 0.051 (0.566) |
| *CC1/2* | 0.996 (0.926) | 0.997 (0.668) |
| Reflections used in refinement | 84,786 | 61,425 |
| Reflections used for R-free | 4239 | 2962 |
| *R*-work | 0.1782 | 0.1915 |
| *R*-free | 0.2189 | 0.2412 |
| Number of non-hydrogen atoms | 9803 | 9140 |
| Macromolecules | 8689 | 8707 |
| Ligands | 18 | |
| Solvent | 1096 | 433 |
| Protein residues | 1040 | 1043 |
| RMS (bonds) (Å) | 0.007 | 0.004 |
| RMS (angles) (°) | 0.76 | 0.57 |
| Ramachandran outliers (%) | 0.00 | 0.00 |
| Average *B*-factor (Å$^2$) | 42.35 | 54.80 |
| Macromolecules | 42.27 | 55.16 |
| Ligands | 53.85 | |
| Solvent | 42.81 | 47.68 |

Statistics for the highest-resolution shell are shown in parentheses.

**Reporting summary**. Further information on research design is available in the Nature Research Reporting Summary linked to this article.

## Data availability

Coordinates and structure factors of CsTOP1-DNA native and selenomethionine forms have been deposited in the Protein Data Bank under accession codes 6Z03 and 6Z01, respectively. Source data are provided with this paper.

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

## Acknowledgements

We are grateful to the core facilities of the Institut Pasteur C2RT, and in particular the scientist from the Crystallography Platform: Ahmed Haouz, Patrick Weber, and Cedric Pissis for performing robot-driven crystallization trials and helpful discussions. We also wish to thank Ariel Mechaly for helping with phasing the SeMet TOP1 structure, and also acknowledge the staff of ESRF and SOLEIL Synchrotrons for assistance and support using beamlines ID29 (Grenoble, France) and SWING and PX2 (Saclay, France). These studies have been supported by a European Research Council (ERC) Grant from the European Union's Seventh Framework Program (FP/2007-2013) (Project EVOMOBIL-ERC) [340440 to P.F.]; l'Agence Nationale de la Recherche [Project ESSPOIR ANR-17-CE12-0032 to T.T., D.G., S.P., and C.M.]. Grants from the Institut Pasteur (Paris), the CNRS (France), and the Université de Paris (France). The studies have also been supported by the Center for Cancer Research, the Intramural Program of the National Cancer Institute, National Institutes of Health [Z01-BC-006161 to K.A., E.K., H.Z., and Y.P.]

## Author contributions

D.G. and T.T. cloned CsTOP1 and constructed the different mutants. E.Y., S.P., T.T., and D.G. produced and purified CsTOP1 proteins. T.T. and S.P. produced crystals and T.T. determined and refined the structure of CsTOP1. E.Y. and D.G. performed the DNA relaxation assays. K.A., H.Z., and E.K. performed TOP1 cleavage reactions. All authors discussed the results. T.T., Y.P., P.F., and C.M. designed the experiments and wrote the paper.

## Competing interests

The authors declare no competing interests.
