## [Peer Review File · Nature Communications]

Topoisomerase I (TOP1) dynamics: conformational transition from open to closed statesREVIEWER COMMENTS

Reviewer #1 (Remarks to the Author):

In this paper Takahashi et al. reported an apo-form topoisomerase I structure from *Caldiarchoaeum subterraneum* (CsTOP1). They claimed that they had identified a 5-residue loop as the hinge between CAP and CAT modules, which permits the relative rotation between the two modules and thus regulates the opening/closing of the enzyme for DNA substrate binding. They also claimed that they had identified a highly conserved tyrosine that plays an important role in the conformational transition of the enzyme. The apo-form structure, which provides an insight into the conformational change associated with DNA binding, is indeed an important addition to the structures of topoisomerase I family, particularly to the type IB subfamily. From crystallographic table and reports, both SeMet-labeled and native structures seem to be solid. However, there are concerns about the presentation of the structure and concerns about some mutagenesis designs and interpretation of experimental data as follows.

1. The whole paragraph starting from the line #27 of page #6 is a copy of the paragraph before it. Please remove it.
2. In DNA relaxation assay, no information about the amounts of enzyme and substrate were given either in text or in figure legend though authors mentioned "the indicated amount of enzyme". A reference to the assay may help.
3. In Figure 4D, the left panel should be a model of CsTOP1 in complex with DNA. However, no details were given about its modeling in any places of the paper and the figure legend of Figure 4D is missing.
4. Authors try to use Figure 4D to suggest that linker domain doesn't bind DNA as observed in all TOP1/DNA complexes is an artifact of molecular packing in crystal. Their suggestion that "in solution, this domain remain close to the DNA molecule in the different steps of the topoisomerase reactions" is very speculative. The linker domain in larger type 1B topoisomerases has not been fully understood yet considering that the linker domain is dispensable in smaller Top 1B molecules.
5. There is no Figure 3C, which has been referred to several times in the paper.
6. Figure 5C seems to be incomplete. It is not clear what authors want to show in this figure.
7. As to the conserved tyrosine (Y207 in CsTOP1), nothing has been done on this residue yet. Authors' statement "this tyrosine is likely critical to guide the motion of the CAP domain upon DNA binding, enabling the closing of the enzyme" is apparently an overshoot. Similar statements also appear in other places of text. It needs to be proved that the conserved tyrosine play such detailed roles.
8. One of the main concerns is about the hinge swap experiment. Authors proposed a 5-residue hinge, which permits relative rotation between CAP and CAT domains. The hinge could be very flexible or rigid in eukaryotic enzyme, and it could accommodate little change in archaeal enzymes as authors mentioned. Based on the structure, the hinge is not on a widely exposed loop. Swapping the 5-residue hinge sequence (WLSDT) with human sequence (MLNPS) could cause direct damages to the protein structure itself. A reduced enzyme efficiency is one of possible results that could be expected. Moreover, it is hard to believe the reduced activity is simply because the hinge becomes less flexible due to a proline in human sequence. Authors can certainly argue that a double mutant (WLSPP) reduced efficiency. Then, the question will be if WLSGG double mutant is going to improve the enzyme activity or not.
9. Another main concern is about the sequence swap experiment at the active site from (LRNYI) to human (KLNYL). The sequence swap is equivalent to a triple mutant at active site. The triple mutation at active site can easily shut down the enzyme. It is absolutely not a way to compare these two enzymes in their local difference in terms of three residues. The rationale of the experimental design is really doubted.
10. In the last sentence of the paragraph starting from line#14 of page#14, authors say "this difference together with linker flexibility, contributes to CPT resistance of the enzyme". There is no knowledge about the linker flexibility of CsTop1. It is not known if it is relatively flexible or relatively rigid.
11. In the first sentence of Discussion, "TOP1 CAP domain freely rotates around the hinge," is

unproved. Please remove the word “freely”.

Reviewer #2 (Remarks to the Author):

The manuscript of Takahashi and coworkers presents the structure of an archaeal type IB topoisomerase. *C. subterraneum* topoisomerase IB (CsTOP1) is interesting as it represents the first example of an archaeal type IB enzyme and also as it allows to compare it with the more widely studied human topoisomerase IB (hsTOPO1). The structure was solved in the absence of DNA and shows an open conformation of the protein, with the N-terminal or CAP domain moved with respect to the rest of the protein. This observation is important as it provides information on the conformation of the protein in the absence of DNA and also on the regions that are responsible for the domain movements; the domains appear to move as rigid bodies with flexible hinges. A few biochemical experiments complement the work although the conclusions are not striking. There are a few points that need to be addressed:

1. The authors disregard comparisons with the smaller type IB enzymes as they are deemed as too different, even though in the final Discussion a brief mention is made. I think disregarding the smaller type IB enzymes is a mistake. They should be included in the comparisons and the movement of the domains in the two cases should be compared. A figure showing the nature of the open conformations in both cases should be included. Comparing with the smaller type IB enzymes does not detract from the major findings of the manuscript and is likely to provide more information and strengthen the conclusions. In particular, it would be interesting to see if the CAP domain moves in similar way in both cases.
2. The authors mention that it may be possible that TOP1 recognizes DNA crossings. This was observed for vaccinia topoisomerase IB by Shuman in 1997 and should be mentioned.
3. The hinge region seems to play a major role; it should be highlighted in the figures by drawing it in a different color. Otherwise, it is very difficult to identify it.
4. The authors write that the hinge residues can be flexible or rigid. It is not clear what they mean by this as it is based on sequence alignments, not direct structural observations. The wording should be clarified.
5. Table II shows the mutants made and the expected effects. It should definitely have a third column showing the actual results. Many of the expected effects were not observed and the third column should state this and a possible reason for it.
6. The sequence alignment in Figure 2 could be moved to the Supplementary section.
7. Figure 3 should have the hinge regions clearly labeled.
8. Figure 4 has an incomplete caption and the panels are too small to see properly. Fewer and larger panels would be more informative; some of the panels can move to the Supplementary section.
9. It is not clear what Figure 5c is trying to show. Is this the correct figure?
10. Figure 6 shows cleavage assays, but without any quantitation. It would be more helpful to show percentage cleavage after proper normalization. For example, in panel c it would help to compare the percentage cleavage with and without CPT.

Overall, this is an interesting manuscript describing important results, but there are some weaknesses. In particular, ignoring previous work on the smaller topoisomerases needs to be changed. Including them is likely to enhance the manuscript and the conclusions. The overall mechanism of relaxation is likely to be very similar for all type IB enzymes and hence including as much available information as possible will strengthen the conclusions and may also help discover other similarities and differences.

Reviewer 1

1. *The whole paragraph starting from the line #27 of page #6 is a copy of the paragraph before it. Please remove it.*

Thank you. We removed this duplication.

2. *In DNA relaxation assay, no information about the amounts of enzyme and substrate were given either in text or in figure legend though authors mentioned “the indicated amount of enzyme”. A reference to the assay may help.*

As suggested, we modified the corresponding material and methods section.

3. *In Figure 4D, the left panel should be a model of CsTOP1 in complex with DNA. However, no details were given about its modeling in any places of the paper and the figure legend of Figure 4D is missing.*

The figure legend of Figure 4D has been completed.

4. *Authors try to use Figure 4D to suggest that linker domain doesn't bind DNA as observed in all TOP1/DNA complexes is an artifact of molecular packing in crystal. Their suggestion that “in solution, this domain remain close to the DNA molecule in the different steps of the topoisomerase reactions” is very speculative. The linker domain in larger type 1B topoisomerases has not been fully understood yet considering that the linker domain is dispensable in smaller Top 1B molecules.*

As suggested, we toned down our statement and wrote "Our structural data highlights the possibility for the linker domain to remain close to the DNA molecule in the different steps of the topoisomerase reaction."

5. *There is no Figure 3C, which has been referred to several times in the paper.*

This was an editorial mistake. Thank you for bringing it up to our attention. Figure 3C in the text has been replaced by Figure 2C.

6. *Figure 5C seems to be incomplete. It is not clear what authors want to show in this figure.*

Figure 5C illustrates the motion of the hinge and how it allows the interaction between one conserved tyrosine and DNA. We modified the figure and the legend to clarify the purpose of this panel.

7. *As to the conserved tyrosine (Y207 in CsTOP1), nothing has been done on this residue yet. Authors' statement “this tyrosine is likely critical to guide the motion of the CAP domain upon DNA binding, enabling the closing of the enzyme” is apparently an overshoot. Similar statements also appear in other places of text. It needs to be proved that the conserved tyrosine plays such detailed roles.*

As suggested, we toned down the sentence and wrote "this tyrosine may be important to guide". Additionally, we performed TOP1 relaxation assay with Y207A mutant and proved that the activity of the enzyme is lost when this residue is mutated (Supplementary Figure 4A). Thank you for the suggestion.

8. *One of the main concerns is about the hinge swap experiment. Authors proposed a 5-residue hinge, which permits relative rotation between CAP and CAT domains. The hinge could be very flexible or rigid in eukaryotic enzyme, and it could accommodate little change in archaeal enzymes as authors mentioned. Based on the structure, the hinge is not on a widely exposed loop. Swapping the 5-residue hinge sequence (WLSDT) with human sequence (MLNPS) could cause direct damages to the protein structure itself. A reduced enzyme efficiency is one of possible results that could be expected. Moreover, it is hard to believe the reduced activity is simply because the hinge becomes less flexible due to a proline in human sequence. Authors can certainly argue that a double mutant (WLSPP)*

reduced efficiency. Then, the question will be if WLSGG double mutant is going to improve the enzyme activity or not.

Thank you for this interesting comment. The full hinge is actually not buried in the structure. While part of the hinge is indeed interacting with the CAP (as indicated in the text), the last two residues are not interacting with the rest of the protein. Thus, we do not believe that there is a direct correlation between the rigidity of the hinge and the activity of the enzyme. As suggested, we performed the relaxation assay with WLSGG and showed a partial loss of activity with this mutant as well (Supplementary Figure 4A).

9. Another main concern is about the sequence swap experiment at the active site from (LRNYI) to human (KLNLYL). The sequence swap is equivalent to a triple mutant at active site. The triple mutation at active site can easily shut down the enzyme. It is absolutely not a way to compare these two enzymes in their local difference in terms of three residues. The rationale of the experimental design is really doubted.

We believe that our presentation was not sufficiently explicit and that Reviewer 1 misunderstood the design and effect of the triple mutant on the enzyme activity. In order to improve the readability of this part, we added a third column in Table 2 compiling the effect of each mutant, and we rewrote this part of the text. As Reviewer 1 explained, it would not be very informative to have a triple mutation at active site that shut down the enzyme. In our assay, we showed the opposite: the triple mutant displays no significant difference with the wild type enzyme in terms of relaxation (Supplementary Figure 4B) and in terms of cleavage (Figure 6C).

10. In the last sentence of the paragraph starting from line#14 of page#14, authors say “this difference together with linker flexibility, contributes to CPT resistance of the enzyme”. There is no knowledge about the linker flexibility of CsTop1. It is not known if it is relatively flexible or relatively rigid.

This is a good point. We do not have direct measure of linker flexibility in our study. Therefore, to avoid overinterpretation, the words "together with linker flexibility" have been removed

11. In the first sentence of Discussion, “TOP1 CAP domain freely rotates around the hinge,” is unproved. Please remove the word “freely”.

The word "freely" was removed. Thank you for your constructive comments.

Reviewer 2

The manuscript of Takahashi and coworkers presents the structure of an archaeal type IB topoisomerase. C. subterraneum topoisomerase IB (CsTOP1) is interesting as it represents the first example of an archaeal type IB enzyme and also as it allows to compare it with the more widely studied human topoisomerase IB (hsTOPO1). The structure was solved in the absence of DNA and shows an open conformation of the protein, with the N-terminal or CAP domain moved with respect to the rest of the protein. This observation is important as it provides information on the conformation of the protein in the absence of DNA and also on the regions that are responsible for the domain movements; the domains appear to move as rigid bodies with flexible hinges.

We appreciate the reviewer comment regarding the importance and novelty of our study.

A few biochemical experiments complement the work although the conclusions are not striking. There are a few points that need to be addressed:

1. The authors disregard comparisons with the smaller type IB enzymes as they are deemed as too different, even though in the final Discussion a brief mention is made. I think disregarding the smaller type IB enzymes is a mistake. They should be included in the comparisons and the movement of the domains in the two cases should be compared. A figure showing the nature of the open conformations

in both cases should be included. Comparing with the smaller type IB enzymes does not detract from the major findings of the manuscript and is likely to provide more information and strengthen the conclusions. In particular, it would be interesting to see if the CAP domain moves in similar way in both cases.

Thank you for this constructive criticism. We agree that smaller type IB enzymes are very interesting and informative in the context of our communication. Accordingly, we have included in or revised manuscript the comparison about the possible internal motion of shorter type IB enzymes.

2. The authors mention that it may be possible that TOP1 recognizes DNA crossings. This was observed for vaccinia topoisomerase IB by Shuman in 1997 and should be mentioned.

Thank you for noticing this unintended oversight. The reference has been included in our revision.

3. The hinge region seems to play a major role; it should be highlighted in the figures by drawing it in a different color. Otherwise, it is very difficult to identify it.

Thank you for the suggestion. We have highlighted the hinge in orange in all figures.

4. The authors write that the hinge residues can be flexible or rigid. It is not clear what they mean by this as it is based on sequence alignments, not direct structural observations. The wording should be clarified.

Good point: we clarified this part to highlight that our observation is based on sequence alignments.

5. Table II shows the mutants made and the expected effects. It should definitely have a third column showing the actual results. Many of the expected effects were not observed and the third column should state this and a possible reason for it.

As suggested, we added a third column to describe the cleavage properties of the mutants. Thank you.

6. The sequence alignment in Figure 2 could be moved to the Supplementary section.

We believe that including the sequence alignment in the main figures is helpful for the reader to readily locate the different residues in the sequences, and to compare human and archaeal enzymes.

7. Figure 3 should have the hinge regions clearly labeled.

We labeled the hinge regions. Thank you.

8. Figure 4 has an incomplete caption and the panels are too small to see properly. Fewer and larger panels would be more informative; some of the panels can move to the Supplementary section.

As suggested, we modified Figure 4 to improve the readability of the panels. We also removed the Figure 4B that was redundant with Figure 3B.

9. It is not clear what Figure 5c is trying to show. Is this the correct figure?

Figure 5C illustrates the motion of the hinge and how it allows the interaction between one conserved tyrosine and DNA. We modified the figure and the legend to clarify the purpose of this panel.

10. Figure 6 shows cleavage assays, but without any quantitation. It would be more helpful to show percentage cleavage after proper normalization. For example, in panel c it would help to compare the percentage cleavage with and without CPT.

As suggested, the quantification of the cleavage bands has been added to figure 6.

Overall, this is an interesting manuscript describing important results, but there are some weaknesses. In particular, ignoring previous work on the smaller topoisomerases needs to be changed. Including

them is likely to enhance the manuscript and the conclusions. The overall mechanism of relaxation is likely to be very similar for all type IB enzymes and hence including as much available information as possible will strengthen the conclusions and may also help discover other similarities and differences.

Thank you.

REVIEWER COMMENTS

Reviewer #1 (Remarks to the Author):

Authors have improved their manuscript, MOSTLY following reviewers' comments. There are still some issues in this revised version to be resolved.

1. There are numerous grammar mistakes through the paper. Some of them cause confusion and it is hard to read. A professional proofreader is STRONGLY recommended to clean it up by improving grammar, punctuation, and syntax, etc.

For example, in Lines 163 to 166 of the Page 8:

The topoisomerase reactions were performed for 0–4 min at 65°C and were stopped by adding 0.5% SD on ice. A total of 2 µl of proteinase K (1 mg/ml) were (was) then added for an additional incubation of 15 min at 55°C. The digestions were (digestion was ?) stopped with the blue charged electrophoretic buffer.

Please define what “blue charged electrophoretic buffer” is. It would be better to use “electrophoresis buffer” instead of “electrophoretic buffer”.

2. In Figure 6(a), the red numbers, 24 and 18, above DNA sequences should indicate HsTop1 cleavage sites while the blue numbers, 11 and 8, indicate CsTop1 cleavage sites. The color scheme notes below the DNA sequences are obviously wrong.

From the gels in Figure 6(b and c), 24-nt DNA fragment is apparently one of two expected (preferred) DNA fragments from the cleavage of HsTop1. The other fragment is 18-nt DNA. However, in the text, the authors consistently say they are 25-nt and 18-nt fragments. Such inconsistencies should not have appeared in a revised manuscript.

3. Authors mentioned “HsTop1 sensitivity to CPT has been shown to be related to the dynamics of TOP1 activity. The DNA rotation rates determine the window of time during which DNA is cleaved by TOP1, the nicked DNA forming the pocket that will accommodate the camptothecin drug.” At the end of section TOP1 dynamics and camptothecin sensitivity, a summary about possible impacts to DNA rotation from mutants is naturally expected. The summary in current version seems to be irrelevant, talking only about the transition between open and closing conformations. They are different steps in the catalytic cycle of the enzyme.

4. The last section of Results, Internal motion in the case of smaller type IB topoisomerase is a new section in this revised version.

There are two paragraphs in this section. The second paragraph is a duplication of the first paragraph. Such a copy-and-paste mistake already happened in other part of the manuscript in the early version. Did anyone ever go through the manuscript before submission?

In fact, the new Results section doesn't provide any new information. Authors could move some discussions from there to the Conclusion.

We are thankful to reviewer #1 for meticulously reviewing our revised manuscript. As suggested, we have carefully edited and further corrected the manuscript.

1. There are numerous grammar mistakes through the paper. Some of them cause confusion and it is hard to read. A professional proofreader is STRONGLY recommended to clean it up by improving grammar, punctuation, and syntax, etc.

For example, in Lines 163 to 166 of the Page 8:

The topoisomerase reactions were performed for 0–4 min at 65°C and were stopped by adding 0.5% SD on ice. A total of 2 µl of proteinase K (1 mg/ml) were (was) then added for an additional incubation of 15 min at 55°C. The digestions were (digestion was?) stopped with the blue charged electrophoretic buffer.

Please define what “blue charged electrophoretic buffer” is. It would be better to use “electrophoresis buffer” instead of “electrophoretic buffer”.

Our answer: Thank you for going over the Materials & Methods and suggesting further editing. As suggested, we have carefully proofread the manuscript, especially on the Material and Methods section to correct the grammatical mistakes. As suggested, we added the composition of the buffers.

2. In Figure 6(a), the red numbers, 24 and 18, above DNA sequences should indicate HsTop1 cleavage sites while the blue numbers, 11 and 8, indicate CsTop1 cleavage sites. The color scheme notes below the DNA sequences are obviously wrong.

From the gels in Figure 6(b and c), 24-nt DNA fragment is apparently one of two expected (preferred) DNA fragments from the cleavage of HsTop1. The other fragment is 18-nt DNA. However, in the text, the authors consistently say they are 25-nt and 18-nt fragments. Such inconsistencies should not have appeared in a revised manuscript.

Our answer: Thank you for noting the error and avoiding this inconsistency.

3. Authors mentioned “HsTop1 sensitivity to CPT has been shown to be related to the dynamics of TOP1 activity. The DNA rotation rates determine the window of time during which DNA is cleaved by TOP1, the nicked DNA forming the pocket that will accommodate the camptothecin drug.”

At the end of section TOP1 dynamics and camptothecin sensitivity, a summary about possible impacts to DNA rotation from mutants is naturally expected. The summary in current version seems to be irrelevant, talking only about the transition between open and closing conformations. They are different steps in the catalytic cycle of the enzyme.

Our answer: Thank you for the suggestion. A short addition has been included to clarify this point. Yet, we avoided to be too speculative, considering that further work would be required to fully understand which molecular dynamic steps are essential for camptothecin sensitivity.

4. The last section of Results, Internal motion in the case of smaller type IB topoisomerase is a new section in this revised version. There are two paragraphs in this section. The second paragraph is a duplication of the first paragraph. Such a copy-and-paste mistake already happened in other part of the manuscript in the early version. Did anyone ever go through the manuscript before submission? In fact, the new Results section doesn't provide any new information. Authors could move some discussions from there to the Conclusion.

Our answer: We are sorry about this duplication and editorial error, which did not appear during the first round of revisions. The duplication has been removed from the manuscript. We decided to keep the last section of the Results in this section because, as suggested by Reviewer #2, it includes novel insights. Thank you.